# Photodynamic Therapy in Pigmented Basal Cell Carcinoma—A Review

**DOI:** 10.3390/biomedicines11113099

**Published:** 2023-11-20

**Authors:** Ewelina Mazur, Dominika Kwiatkowska, Adam Reich

**Affiliations:** 1Department of Dermatology, Institute of Medical Sciences, Medical College of Rzeszow University, 35-055 Rzeszow, Polanddominika.kwiatkowska1808@gmail.com (D.K.); 2Doctoral School, University of Rzeszow, 35-959 Rzeszow, Poland

**Keywords:** basal cell carcinoma, BCC, photodynamic therapy, PDT, pigmented tumors

## Abstract

This review summarizes the effectiveness of photodynamic therapy (PDT) in the treatment of the pigmented subtype of basal cell carcinoma (BCC) based on the current literature. PDT is a light-activated treatment, non-invasive, that selectively destroys tumor cells and tissues via the interaction of a photosensitizer, light, and molecular oxygen. It can induce cancer cell death through direct tumor vascular damage or via the induction of immune response. However, human skin is also an absorption and scattering medium since it contains hemoglobin and melanin that act as chromophores. Eumelanin can be considered a light-absorber and an intracellular antioxidant that can neutralize PDT-induced ROS and, therefore, decrease PDT success. Various factors, including tumor depth, the degree of pigmentation in malignant cells, and the individual’s skin phototype, can impact the outcome of this intricate biochemical process. It has been widely recognized that PDT exhibits limited efficacy in the treatment of pigmented lesions. However, new combination techniques such as curettage or debulking before PDT show promising results in the treatment of pigmented BCC.

## 1. Introduction

Basal cell carcinoma (BCC) is the most frequent skin cancer, especially in the white skin population, with the incidence rate still increasing worldwide [1]. The most common clinical variants of BCC are nodular and superficial ones. Each of them may present pigmented structures and, thus, become so-called pigmented BCC [2]. As BCC is characterized by a slow progression, rare formation of distinct metastases, and low mortality rate, the most challenging aspect of managing BCC remains the risk of local tumor recurrence. Although surgery is considered the gold standard for invasive subtypes of BCCs, in the case of non-invasive subtypes, many alternative treatment modalities have been introduced throughout the years to limit the disfigurement risk [3]. These methods encompass cryotherapy, curettage, electrodesiccation, topical drugs, and, last but not least, photodynamic therapy (PDT) [4]. 

PDT is a light-activated treatment modality that is non-invasive and selectively destroys cells and tissues via the interaction of a photosensitizer, light, and molecular oxygen. The excited photosensitizer (PS), through an intersystem crossing process, can undergo transformation into a long-lived excited triplet state and launch two kinds of photochemical reactions with adjacent molecules. In the type I photochemical reaction, PS interacts directly with cell membrane polyunsaturated fatty acids and forms organic radicals that generate cytotoxic reactive oxygen species (ROS) and launch free radical chain reactions. The type II photochemical reaction results in the formation of a powerful oxidizing agent, singlet oxygen (1O2), via energy transfer to molecular oxygen [5]. In addition, recent studies documented another important mode of PDT action that impedes mitochondrial electron flow, namely, tandem photocatalysis [6,7]. This process, alternatively to the more classic type II photosensitization pathway, is useful within oxygen-depleted environments.

PDT acts through direct tumor death, vascular damage, and activation of the immune response. First, it leads to direct photodestruction of malignant cells, then to ROS-generated vascular damage, and finally enhanced (T-cell dependent) immunological recognition. Different cell death pathways that mediate the cytotoxicity of PDT may play a role including necrosis, apoptosis, and autophagy. However, non-conventional cell death modalities can also be induced by PDT, such as mitotic catastrophe, paraptosis, pyroptosis, and parthanatos, as well as forms of regulated necrosis such as necroptosis, and ferroptosis (Figure 1) [8,9]. The balance of these cell death mechanisms is driven largely by the subcellular localization inherent to a specific photosensitizer and its incubation conditions, as well as the light dose. Therefore, new compounds and innovative methods are still being researched to improve PDT outcomes, especially, to successfully treat deeper lesions. Several PDT modifications have been reported including repeated procedures, new lipophilic compounds, third-generation PS, debulking, laser-assist, and others [10]. However, melanin, which is a natural pigment in the skin, is a known factor decreasing PDT efficacy independently of tumor thickness [11,12]. Interestingly, data aiming to counter the pigmentation in pigmented basal cell carcinoma (pBCC) PDT treatment procedures are limited. Therefore, we have performed a literature review to summarize the current knowledge on the effectiveness of PDT in the treatment of the pigmented subtype of BCC. 

## 2. Melanin in Photodynamic Therapy

The basal layer of the epidermis contains melanocytes (MCs). These cells are responsible for skin pigmentation and protection against UV radiation via melanin formation [13]. Melanin is enzymatically synthesized at approximately 10-nm granular sites that stud the internal walls of organelles known as melanosomes. Melanosomes may contain a variable amount of melanin, i.e., cutaneous melanosomes may contain 1/4th to 1/10th of the melanin concentration of the melanosomes found in the retinal pigmented epithelium. 

### 2.1. Melanin as a Light-Absorber

From an optical point of view, human skin is an absorption and scattering medium since it contains hemoglobin (oxygenated and deoxygenated) and melanin which can be considered as chromophores that show distinctive optical absorption properties within the visible wavelength range [14]. 

The average epidermal absorption coefficient (μa) depends on both the melanosomal μa (1.7 × 1012 nm–3.48 [cm^−1^] for skin, where nm equals the wavelength in nm) and the volume fraction (fv) of melanosomes within the epidermis. Fv varies depending on the skin color from 1% in pale skin to 5% in darker skin [15]. Some authors, on the other hand, claimed that fv for light-skinned Caucasians equals 1–3%, for well-tanned Caucasians and Mediterraneans 11–16%, and for darkly pigmented Africans 18–43% [16]. 

According to Shimojo et al., both μa ethnic differences and reduced scattering coefficients (μ’s) have a limited effect on the depth of the PDT treatment. At the same time, the differences present in energy deposition (S) can cause great variances in the production of heat in Asian, Caucasian, and African epidermises. The difference in the values of the μa in different skin types affects the S of the epidermis. At 700 nm, the epidermal layer is characterized by the maximum S. In Caucasian skin tissue, S is two times lower than that in Asian and four times lower than that in African skin tissue [17]. Moreover, the μa of sun-protected skin is generally lower than that of sun-exposed areas within each skin type group in the wavelength range from 500 to 900 nm. Furthermore, the μa spectra of sun-exposed skin type V-VI have a greater slope in the 600–800 nm wavelength region compared to the skin types I-II and III-IV. The average μ′s of the skin type V-VI group is 1–10% higher than those of the skin type I-II and III-IV groups. This phenomenon may be due to the fact that the density of scatterers in the skin, such as melanin, is higher in the skin type V-VI group than in skin type I-II and III-IV groups [18]. 

Photosensitizers used for PDT to treat dermatologic conditions, such as 5-aminolevulinic acid (5-ALA) and its methyl ester (methyl aminolevulinate—MAL), are mainly metabolized into protoporphyrin IX (PpIX), which has five absorption wavelength peaks: 410 nm, 510 nm, 545 nm, 580 nm, and 630 nm [19]. Red light (630–700 nm), most commonly used in PDT, is known to have the deepest penetration depth, however, the presence of melanin in a stable protein complex with a wide absorption spectrum in the same tissue, competes with PS for photons resulting in inefficient phototoxicity. However, in the typical skin melanin absorption spectrum, only a small peak exists beyond the 650 nm wavelength. Generally, the absorbance values of melanin rise gradually from 750 nm to 600 nm, then moderately from 600 nm to 450 nm, and finally rise sharply from 450 nm to a broad peak at 335 nm [20,21,22]. Therefore, in recent years, highly active PS absorbing or primed in the near-infrared spectral region (700–900 nm) that might allow the use of PDT even in highly pigmented lesions is being developed [23,24,25]. 

### 2.2. Melanin as an Intracellular Antioxidant

Melanins are pigments of high molecular weight formed by oxidative polymerization of phenolic or indolic compounds [26]. Melanin formation starts with the oxidation of tyrosine to dopaquinone. Later, dopaquinone can either undergo cyclization leading, after a further oxidation (to 5,6-dihydroxyindole (DHI) or 5,6-dihydroxyindole-2-carboxylic acid (DHICA)) and polymerization, to the creation of melanin pigments brown or dark in color, known as eumelanins. On the other hand, dopaquinone can become entrapped by cysteine (creating benzothiazine (BT) intermediates and benzothiazole (BZ)) and polymerize to synthesize the reddish-brown pigments known as pheomelanins [27]. The human epidermis comprises approximately 76% of eumelanin and 24% pheomelanin (including the 4 moieties ratio: DHI 35%, DHICA 41%, BZ 20%, and BT 4%), regardless of the degree of pigmentation (Figure 2). However, lighter skin phototypes possess low content of photoprotective eumelanin thereby explaining the higher sensitivity toward UV exposure [28,29]. 

Eumelanin can be considered an intracellular antioxidant that can neutralize PDT-induced ROS and decrease PDT success [30]. DHICA-melanin possesses higher antioxidant activity when compared to DHI-melanin. This phenomenon is attributed to its poorly aggregated structure [27,31]. It can act as an efficient antioxidant and hydroxyl radical scavenger, as well as an inhibitor of lipid peroxidation, and can protect the tissue against hydrogen peroxide-induced cytotoxicity via activation of the Nrf-2 pathway [32,33,34,35,36,37]. Contrarily, pheomelanin exerts photosensitizing characteristics and leads to the UV-induced production of ROS [38]. In the study by Tanaka et al. on UVA-exposed pheomelanin, BT and BZ monomers showed similar pro-oxidant activities. However, the effects of ROS scavengers exhibited a large difference where BZ monomers were more reactive than BT monomers. Probably, the redox reactions in BZ monomers may proceed via singlet oxygen and in BT monomers via superoxide anions [39,40]. 

Lawrence et al. demonstrated that wavelengths of 385–405 nm can cause dark cyclobutane pyrimidine dimer (CPD) formation and, therefore, lead to significant damage, both in vitro and in vivo. This phenomenon is likely caused by oxidative stress generated by chromophores in the skin that absorb strongly in this region, such as melanins, PpIX, and β-carotene [41]. CPDs are the most frequent DNA changes responsible for ultraviolet (UV) carcinogenesis [42,43,44]. PDT-induced ROS, however, does not lead to CPD generation in melanin-containing cells [45]. PDT locally generates the intracellular ROS dependent on photosensitizer localization, whereas, in UVA radiation, ROS is generated throughout the whole cell [46]. Additionally, UVA-ROS-induced CPD requires nuclear melanin, while PDT does not result in the formation of melanin monomers and nuclear transport [47]. 

PDT is known to exert a whitening effect in vitro and in vivo by reducing melanin content and tyrosinase activity [48]. Melanogenesis can also be inhibited via a paracrine effect when melanocytes are exposed to PDT-treated keratinocytes or dermal fibroblasts. This is likely due to a decreased release of Kit ligand and hepatocyte growth factor which are melanocyte-stimulating cytokines [49]. Furthermore, PDT reduces mottled hyperpigmentation of photoaged patient skin [50,51]. However, some authors found that in heavily pigmented lesions, an increase in the amount of melanin can be observed. This is probably due to the auto-oxidation of melanin precursors present in the lesions via free radicals and de novo synthesis of melanin triggered by PDT. Nevertheless, the increase in melanogenesis does not protect against the PDT-induced DNA and cytoskeleton damage [52]. The hyperpigmentation induced by PDT is more pronounced in skin types III and above [53].

## 3. Photodynamic Therapy in Pigmented Basal Cell Carcinoma

The literature search followed PRISMA guidelines for systematic reviews [54] and the Cochrane manual [55]. Sources included the PubMed (MEDLINE) and Scopus databases. The initial search was complemented by a manual search of reference lists from retrieved articles. The following search strategy was used: (photodynamic therapy) OR (PDT) AND (basal cell) OR (bcc) AND (melanin) OR (pigment). The search yielded 482 results. Where relevant, articles were read in full, and then the decision about the inclusion of an article was made. A total of 473 references were not relevant to the scope of the review because they did not entail cases of BCCs treated with PDT or did not refer to pigmented variants of BCC. A total of nine studies met the final inclusion eligibility and were included in the review. 

Characteristics of pigmented BCC lesions treated with PDT are summarized in Table 1. 

### 3.1. PDT in pBCC without Prior Debulking

In 2005, Kaviani et al. treated 30 cases of BCC, 18 of them pigmented, with PDT. They observed that the pBCCs response rate to a single PDT session was significantly lower than the other subtypes, as it was equal to 14% while it was 100%, 90%, and 62% for ulcerative, nodular, and superficial BCCs, respectively. They suggested that the pigment within the pBCCs may prevent adequate light absorption, hindering the PDT effect in those lesions [56]. 

Ramirez et al. treated sixteen pBCC lesions with MAL-PDT in two treatment sessions [57]. They obtained a complete response in 50% of treated lesions and a partial response in another 50% of the cases. Ramirez et al. observed that the pigmented region of the BCC absorbs both excitation as well as fluorescence emission from PpIX. According to Ramirez et al., these absorbing regions created a shielding effect during irradiation, reducing the light dosage to the lesion and resulting in a lack of PDT response at the tumor’s deep margin. They suggested that similar to nodular BCCs, pBCCs might benefit from prior curettage and/or debulking. The success of this procedure might, however, be greatly dependent on the skills of the physician performing the procedure [57].

In a case report by Lin et al., the success of the sequential use of etretinate and PDT in treating two keratotic, pigmented nodular BCCs was described [58]. A 2-month course of oral etretinate at a dose of 0.5 mg/kg per day was used to decrease superficial scaling. A double PDT was performed on days 0, 3, and 5 with the use of a 2% ALA solution for 16 h before irradiation. These authors were the first ones who used the double irradiation PDT scheme, which was dictated by the fact that the red fluorescence of PpIX initially disappeared after the first PDT irradiation but reappeared 90 min later. It implied a new synthesis of PpIX by deeper, still viable tumor cells [58].

### 3.2. PDT in pBCC with Prior Debulking/Curettage

In 2007, Souza et al. treated eighteen lesions in six patients with purified hematoporphyrin derivatives (PHD)-PDT. Two of those 18 lesions were pigmented (one superficial and one nodular subtype). Twelve lesions were treated with PDT alone, while six nodular lesions were selected for prior debulking. The nodular lesion presented with complete response (only superficial pigmentation along the scar border), while the superficial lesion presented with partial response to the treatment. Therefore, the procedure of prior curettage of primary BCC may help to increase the cure rate by either reducing the tumor mass prior to implementation of intravenous PS or improving the transdermal penetration of topical PDT agents [59]. 

The Japanese group, led by Itoh, treated sixteen pBCCs (eleven nodular, two superficial, and three ulcerative) located in the head and neck area [60]. The PDT procedure was preceded by electro-curettage under local anesthesia. This resulted in the removal of visible pigmentation (either partially or completely) and reduced the tumor volume. Immediately after electro-curettage, ALA-PDT was performed. This procedure was carried out at two or three-week intervals three or more times (curettage was stopped when the lesion’s pigmentation became visibly undetectable). In total, fourteen out of sixteen pBCCs showed complete response (two nodular BCCs showing partial or no response were excised). No recurrences were seen up to 6 months after treatment. The additional benefit of the combination of electro-curettage with ALA-PDT was a smaller post-surgery scar compared to standard surgical procedures such as Mohs micrographic surgery [60]. These authors also mentioned that in their previous attempts they tried, with little result, bleaching the pigmentation in a pBCC using a normal-mode ruby laser and Q-switch YAG laser with high power [60].

A Spanish group led by Garcia-Cazana presented a response rate in pBCCs treated with PDT of 76.19% [61]. These results were almost identical to nodular BCCs (76.8%), for which PDT is indicated. However, at the same time, the recurrence rate was established at about 19%. They treated 25 lesions (66.7% in head and neck area) with MAL-PDT. The session was preceded by curettage debulking, up to the point where the macroscopic pigment was completely removed [61]. 

Pereyra-Rodrizguez et al. [62] presented a case in which five pigmented facial BCCs (three superficial and two nodular) were treated in the same patient with a standard regimen of MAL-PDT on day 0 and 7 days after the initial procedure. Superficial lesions were scraped to increase penetration of the PS, whereas nodular lesions were shaved over the tumor margins. All lesions achieved a 100% response, and there were no recurrences at the 12-month follow-up [62].

In 2021, Salvio et al. published an article regarding the long-term follow-up among patients with multiple pBCCs [63]. This study included two patients with a total of thirty lesions. The pBCC lesions were debulked (removal of the entire tissue above the skin level with a blade), and then a standard MAL-PDT procedure was performed. Immediately after the first illumination, the whole procedure was repeated. However, this time the MAL incubation period was shortened to 90 min. All treated lesions showed a histologically confirmed complete response 30 days after treatment. The clinical follow-up showed no recurrence for all lesions with a mean follow-up time of 24 months. They summarized that as the visible pigmented area was removed, the remaining microscopic pigment probably did not affect the efficacy of PDT [63]. 

### 3.3. PDT in pBCC with Prior Fractional CO_2_ Laser

Sung et al. ablated the epidermis of three histologically confirmed pBCCs using a fractional CO_2_ laser (tip size of 120 μm, peak power of 30 W, pulse energy of 50 mJ, and 200 spots/cm^2^) [64]. Next, a 16% MAL-PDT was performed with a shortened incubation time (90 min), a light wavelength of 630 nm, and a light dose of 37 J/cm^2^. Treatment was repeated in four-week intervals until it was no longer palpable and no pigmentation could be observed. All patients achieved complete resolution of pBCCs in an average of four treatment sessions. Interestingly, despite the full-thickness ablation of the epidermis before PDT, they noted no scar formation. Therefore, PDT may have affected the healing process of the ablated tissue [64].

## 4. Other Photodynamic Therapy Resistance Factors of Basal Cell Carcinomas 

PpIX formation in human skin declines in an age-dependent manner. Nissen et al. found that MAL induced a higher PpIX fluorescence in nondysplastic young skin than in the dysplastic skin of elderly patients [65], a surprising phenomenon because MAL is believed to have a higher affinity for dysplastic lesions [66]. Due to the more lipophilic structure, MAL is believed to have an enhanced penetration through the stratum corneum compared to ALA. However, new nanoformulations of ALA induce more PpIX formation than MAL, suggesting its superior transdermal delivery [65].

In 2017, Garcia-Cazana et al. reported that BCC response to MAL-PDT was independent of age, sex, and both the size and location of the tumor. Although the difference was not statistically significant, the mean age of non-responders was higher, probably due to the age-associated decline in PpIX formation in human skin [61]. In 2019, the same group decided to identify potential biomarkers of BCC responses to MAL-PDT. To this end, they analyzed skin samples from patients with BCC treated with MAL-PDT and also studied the effects of MAL-PDT in two representative BCC murine cell lines. The mean age of non-responders was higher (74.36 years) than in responders (69.22 years) (*p* = 0.007). The authors found that higher response rates were observed in superficial BCCs when compared with nodular BCCs (87.5% vs. 74.5%; *p* = 0.487). The poorest response rate was seen in BCCs located on the nose (62.7%), while the best response rate was for tumors located on the trunk (94.7%) (*p* = 0.003). Moreover, the cure rate was higher for lighter versus darker phototypes (89.1% vs. 66.7%; *p* = 0.034). Of the histological variables, only peritumoral inflammatory infiltrate determined a higher response rate (85.7%; *p* = 0.032). Immunohistochemical variables with a statistically significant association with the response to MAL-PDT were positive p53 immunostaining (observed in 84.6% of responders but only 15.4% of non-responders; *p* = 0.011) and β-catenin immunostaining (moderate or intense expression in 84.6% of responders and in 33.3% of non-responders; *p* = 0.096) with patterns of peripheral reinforcement of basaloid islands [67].

Dermoscopy is a tool helpful in predicting response to therapy or assisting the tumor response to different treatments [68]. It can also help identify structures associated with BCCs’ lower response to MAL-PDT. They, as expected, encompass superficial pigmented structures located at the both dermal-epidermal junction as well as the superficial papillary dermis (“spoke wheels”, “concentric structures”, “leaf-like areas”) as well as deeper pigment structures (blue globules, blue “ovoid nests”) [69].

## 5. Future Perspectives

The laser-assisted PDT has gained more interest in recent years. Ablative laser pretreatment disrupts the stratum corneum and on its own adds a therapeutic effect. Several preclinical studies showed that pretreatment with a laser facilitated the accumulation of topical drug formulations, as well as reduced photosensitizer incubation time [70,71]. In the article by Genouw et al., 12 months after treatment, a 100% efficacy was achieved for superficial BCCs in a group treated with a continuous CO_2_ laser plus PDT or fractional CO_2_ laser plus PDT [72]. However, only one case series in pBCCs was performed by using this method [64]. Thus, further studies are definitely needed. 

Currently used photosensitizers (5-ALA, MAL) are characterized by limited skin penetration and inferior luminescence. Therefore, new delivery systems such as microvehicles, nanoparticles, nanoliposomes, and micelles are developed with promising results even in malignant melanoma [73,74]. Similar studies are needed to enhance the PDT treatment efficacy in PBC.

Moreover, as we enter the era of personalized medicine, research should focus on factors contributing to patient resistance to treatment. This would allow physicians to tailor a specific treatment mode to an individual patient and make them benefit to the fullest from the proposed therapy.

A summary of factors influencing PDT success in pBCC is presented in Figure 3.

## 6. Conclusions 

Taking the above-mentioned facts into consideration, the PDT effect should be more prominent in patients with fair skin and on sun-protected, thin epidermis. Melanin content could be the main driver for decision-making processes. Moreover, a light length of 635 nm, mostly used in PDT, is characterized by a small melanin absorption, which should not pose a significant problem for the treatment effects. Despite the fact that PDT is thought to increase melanogenesis, it does not protect target lesions from DNA and cytoskeleton damage, probably through the keratinocyte paracrine effect. A combination of tumor debulking by either electro-curettage, standard scalpel removal, or laser ablation, increases the success rates of ongoing pBCC therapy. Likewise, procedures using double irradiation in one sitting or repeated techniques (1 or 2 weeks apart), are characterized by higher complete response rates. However, face lesions, especially eyelids, might not be good candidates for PDT, as they achieved the lowest treatment response. Patients with higher phototypes might require higher total light doses and repeated procedures. Dermoscopy is a helpful tool in predicting the tumor response to undergoing treatment. It can also assist BCC debulking techniques by identifying features characteristic of persistent pigment structures.

Nevertheless, there is a need for better-quality studies regarding the use of PDT in pBCCs. Future research should compare existing protocols in a randomized, blinded fashion. Moreover, it should include clinical and/or dermoscopic features that enable classification of the lesions as pigmented ones (by describing specific structures and subtypes) as well as determining the degree of pigmentation. For each lesion, Fitzpatrick skin phototype should be mentioned to assess skin type influence on overall treatment response. Unfortunately, to date, there are no studies that have assessed the response rate to PDT in different Fitzpatrick skin phototypes. New non-invasive diagnostic tools (such as skin ultrasound, confocal microscopy, or optical coherence tomography) can be used to better assess tumor depth and find features predictive of treatment failure or success. The development and use of new PS that specifically target pBCC cells, together with studies regarding patient-specific treatment resistance factors, should be prompted.

## Figures and Tables

**Figure 1 biomedicines-11-03099-f001:**
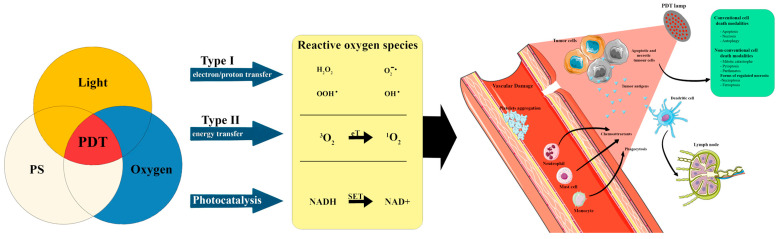
Schematic illustration of photodynamic reactions (type I, type II, and photocatalysis) and cell death pathways in the process of photodynamic therapy (PDT—photodynamic therapy; PS—photosensitizer; eT—electron transfer; SET—single electron transfer; 1O2—singlet excited state; 3O2—triplet excited state). The Figure was partly generated using Servier Medical Art, provided by Servier, licensed under a Creative Commons Attribution 3.0 unported license.

**Figure 2 biomedicines-11-03099-f002:**
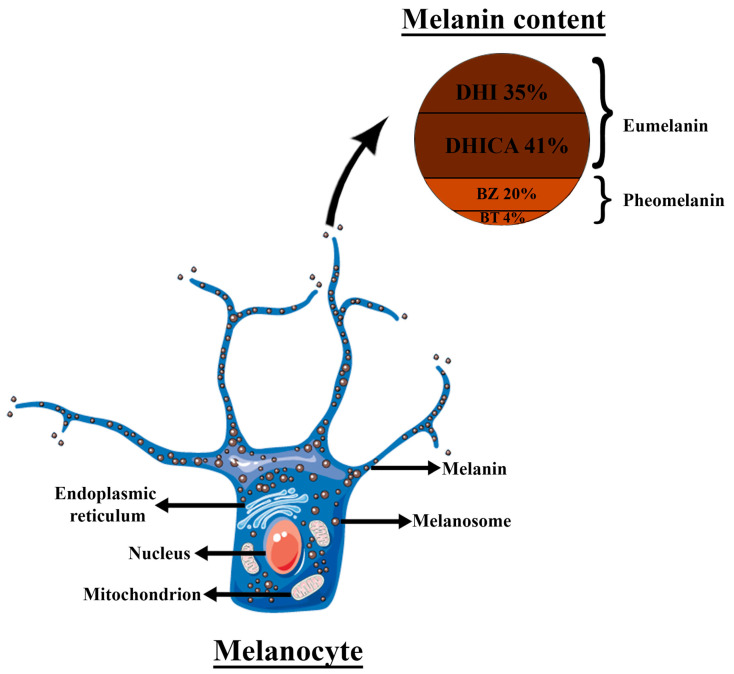
Schematic illustration of melanocyte structure and melanin granule content of the human epidermis; DHI—5,6-dihydroxyindole; DHICA—5,6-dihydroxyindole-2-carboxylic acid; BT—benzothiazine; BZ—benzothiazole. The Figure was partly generated using Servier Medical Art, provided by Servier, licensed under a Creative Commons Attribution 3.0 unported license.

**Figure 3 biomedicines-11-03099-f003:**
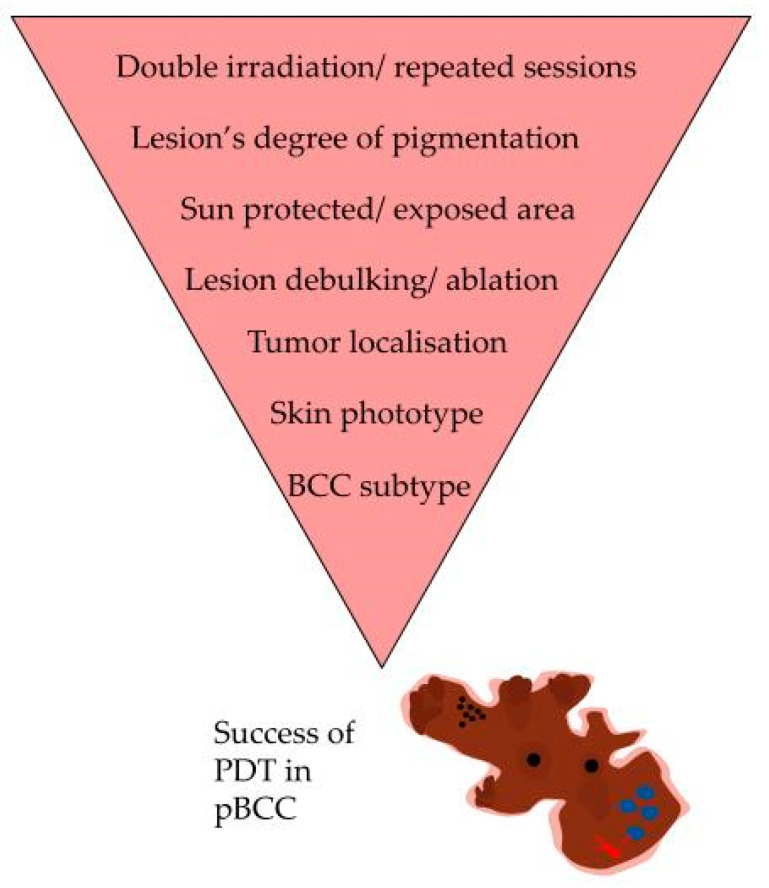
Schematic representation of pigmented basal cell carcinoma and factors influencing PDT success in this indication. PDT—photodynamic therapy; pBCC—pigmented basal cell carcinoma. The Figure was partly generated using Servier Medical Art, provided by Servier, licensed under a Creative Commons Attribution 3.0 unported license.

**Table 1 biomedicines-11-03099-t001:** Characteristics of pigmented basal cell carcinoma lesions treated with photodynamic therapy.

Study	Patient Number	Sex/Age	Lesion Number	Tumor Localisation	Tumor Subtype	PS	Number of PDT Sessions	Light Dose per PDT Session	Debulking/Curettage	Light Length	Response Rate
Kaviani et al. (2004); [56]	1	M/47	1–7	cheek 1 8vertex 2 8nose 3 8glabella 1	NS	PHD	1	200 J/cm^2^	no	632 nm	no response
	2	M/65	1–9	frontal 1 8temporal 2 8vertex 1 8neck 1 8occipital 2 8parietal 2	NS	PHD	1	100 J/cm^2^	no	632 nm	no response
			10	retro-auricular	NS	PHD	1	100 J/cm^2^	no	632 nm	partial response (40–74%)
Ramirez et al. (2014); [57]	NS	NS	1–16	NS	NS	MAL 20%	2	150 J/cm^2^	yes (surface debridement (in sBCC, curettage without local anesthesia for nBCC)	630 ± 10 nm	complete response in 50% and partial response in another 50% of the cases
Lin et al. (2009); [58]	4	F/68	1–2	scalp	Nodular	2% ALA solution	3	120 J/cm^2^	2-month course of oral etretinate 0.5 mg⁄kg per day	630 ± 40 nm	complete response
Souza et al. (2007); [59]	5	F/75	1	scalp	nodular-ulcerative	PHD	1	300 J/cm^2^	curettage	630 nm	complete response *
			2	temporal	superficial	PHD	1	300 J/cm^2^	no	630 nm	partial response
Itoh et al. (2000); [60]	6	F/60	1	right inner canthus	nodular	10% ALA instillation plus 20% ALA emulsion	3	100–500 J/cm^2^	electro-curettage of pigmentation under local anesthesia	630 nm	complete response
	7	M/75	1	nose	superficial	10% ALA instillation plus 20% ALA emulsion	3	100–500 J/cm^2^	electro-curettage of pigmentation under local anesthesia	630 nm	compete response
	8	F/71	1	right lower eyelid	nodular	10% ALA instillation plus 20% ALA emulsion	3	100–500 J/cm^2^	electro-curettage of pigmentation under local anesthesia	630 nm	complete response
	9	F/79	1	head	superficial	10% ALA instillation plus 20% ALA emulsion	5	100–500 J/cm^2^	electro-curettage of pigmentation under local anesthesia	630 nm	compete response
	10	M/65	1	right nasolabial fold	nodular	10% ALA instillation plus 20% ALA emulsion	4	100–500 J/cm^2^	electro-curettage of pigmentation under local anesthesia	630 nm	complete response
	11	F/56	1	nose	ulcerative	10% ALA instillation plus 20% ALA emulsion	3	100–500 J/cm^2^	electro-curettage of pigmentation under local anesthesia	630 nm	compete response
	12	F/78	1	head	nodular	10% ALA instillation plus 20% ALA emulsion	3	100–500 J/cm^2^	electro-curettage of pigmentation under local anesthesia	630 nm	complete response
	13	M/76	1	right lower eyelid	nodular	10% ALA instillation plus 20% ALA emulsion	3	100–500 J/cm^2^	electro-curettage of pigmentation under local anesthesia	630 nm	partial response
	14	M/77	1	right upper eyelid	nodular	10% ALA instillation plus 20% ALA emulsion	4	100–500 J/cm^2^	electro-curettage of pigmentation under local anesthesia	630 nm	no response
	15	M/73	1	left ala nasi	nodular	10% ALA instillation plus 20% ALA emulsion	3	100–500 J/cm^2^	electro-curettage of pigmentation under local anesthesia	630 nm	complete response
	16	M/69	1	left auricle	ulcerative	10% ALA instillation plus 20% ALA emulsion	5	100–500 J/cm^2^	electro-curettage of pigmentation under local anesthesia	630 nm	complete response
Ramirez et al. (2014); [57]	17	F/76	1	left auricle	nodular	10% ALA instillation plus 20% ALA emulsion	4	100–500 J/cm^2^	electro-curettage of pigmentation under local anesthesia	630 nm	complete response
			2	left auricle	nodular	10% ALA instillation plus 20% ALA emulsion	3	100–500 J/cm^2^	electro-curettage of pigmentation under local anesthesia	630 nm	complete response
	18	F/64	1	nose tip	nodular	10% ALA instillation plus 20% ALA emulsion	4	100–500 J/cm^2^	electro-curettage of pigmentation under local anesthesia	630 nm	complete response
	19	M/85	1	nose	ulcerative	10% ALA instillation plus 20% ALA emulsion	3	100–500 J/cm^2^	electro-curettage of pigmentation under local anesthesia	630 nm	complete response
	20	M/65	1	left ala nasi	nodular	10% ALA instillation plus 20% ALA emulsion	3	100–500 J/cm^2^	electro-curettage of pigmentation under local anesthesia	630 nm	complete response
Garcia-Cazana et al. (2017) #; [61]	# 21–41	# 12M and 9F aged 40–100 (mean 73.05)	# 1–21	# Nose 1 Cheek 1 Forehead 2 Scalp 5 8Ear 3 8Neck 2 8Back 2 8Chest 5	NS	16% MAL	2–3	37 J/cm^2^	curettage with local anesthesia when deeper pigmentation	630 nm	complete response in 76.2% of treated lesions
Pereyra-Rodriguez et al. (2009); [62]	42	F/79	1–2	temporal	Nodular diffusely pigmented	16% MAL	2	37 J/cm^2^	blade debulking	630 nm	complete response
	3	F/71	3–4	frontal 1 8cheek 1	Superficial partially pigmented	16% MAL	2	37 J/cm^2^	debridement	630 nm	complete response
	4	F/79	5	cheek	Nodular partially pigmented	16% MAL	2	37 J/cm^2^	blade debulking	630 nm	complete response
Salvio et al. (2021); [63]	43	F/56	1–13	upper limb 8 8trunk 6	Nodular	20% MAL	2	150 J/cm^2^	blade debulking	630 nm	complete response
	6	F/56	14–20	trunk	Superficial	20% MAL	2	150 J/cm^2^	blade debulking	630 nm	complete response
	44	F/52	1–11	upper limb 8 8trunk 3	Nodular	20% MAL	2	150 J/cm^2^	blade debulking	630 nm	complete response
Sung et al. (2017); [64]	45	F/80	1	thigh	Superficial	16% MAL	3	37 J/cm^2^	fractional CO_2_ laser	630 nm	complete resolution
	46	M/59	1	back	Superficial	16% MAL	4	37 J/cm^2^	fractional CO_2_ laser	630 nm	complete resolution
	47	M/66	1	shoulder	Superficial	16% MAL	5	37 J/cm^2^	fractional CO_2_ laser	630 nm	complete resolution

Abbreviations: ALA—aminolevulinic acid; MAL—methyl aminolevulinate; NS—not specified; PHD—purified hematoporphyrin derivatives; PS—photosensitizer. * After the second session with ALA-PDT, # a study by Garcia-Cazana et al. [61] had no individual lesion/patient information, data included in this Table refers to the whole study population [62].

## Data Availability

The datasets generated during and/or analyzed during the current study are available from the corresponding author upon reasonable request.

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
