# Peer review of "Photodynamic Therapy in Pigmented Basal Cell Carcinoma—A Review"

_biomedicines, 2023, doi:10.3390/biomedicines11113099_

Round 1
Reviewer 1 Report
Comments and Suggestions for Authors
The presented review article provides an overview of PDT in treatment of BCC. I believe that the study is important, and would be of interest to the readers of the journal. Only some minor comments from my side:
1- Figure 1: Please either change the color of the arrows in the right part of the figure or the color of the writing within the arrows, since the text is not clear enough.
2- The title of table 1 is not representative of the presented data and needs to be rephrased, since the authors didn't focus on the clinical outcome of the treatment in the table.
3- In the future perspectives section, the authors need to highlight the current novel direction of loading PS within nanoparticles, to enhance the efficacy of PDT in treatment of BCC.
Author Response
Reviewer: The presented review article provides an overview of PDT in the treatment of BCC. I believe that the study is important, and would be of interest to the readers of the journal.
Authors: We would like to thank the reviewer for the positive comments.
Reviewer: Only some minor comments from my side:
1- Figure 1: Please either change the color of the arrows in the right part of the figure or the color of the writing within the arrows, since the text is not clear enough.
Authors: Modified as requested.
Reviewer: 2- The title of table 1 is not representative of the presented data and needs to be rephrased, since the authors did not focus on the clinical outcome of the treatment in the table.
Authors: The title of the table was modified.
Reviewer: 3- In the future perspectives section, the authors need to highlight the current novel direction of loading PS within nanoparticles, to enhance the efficacy of PDT in treatment of BCC.
Authors: This part has been updated as requested
Reviewer 2 Report
Comments and Suggestions for Authors
The article contains important information regarding the effectiveness of photodynamic therapy (PDT) in the treatment of basal cell carcinomas (pBCCs) and factors influencing therapy outcomes. The authors review existing research and conclude that PDT may be more effective in patients with fair skin and sun-protected skin lesions. They also discuss the impact of melanin on PDT effectiveness and the choice of the appropriate wavelength of light.
The article emphasizes the importance of tumor debulking before PDT and the introduction of new strategies, such as double irradiation in a single session or repeated procedures at weekly intervals. The authors also analyze the significance of the Fitzpatrick skin phototype and diagnostic tools in assessing therapy effectiveness.
The article's conclusions highlight the need for further research to provide more precise guidelines for PDT in treating pBCCs. Special emphasis should be placed on research quality and the use of diagnostic tools for a more accurate assessment of tumor changes.
The entire article provides significant information for specialists involved in pBCC treatment and underscores the importance of further research in this field. Here are some minor comments to improve the article's quality:
Abstract:
It's advisable to avoid introductions like "This scientific article aims to..." and instead directly discuss the article's substance.
In the sentence "PDT is a non-invasive treatment modality," it's recommended to add that PDT is...
In the sentence "Different factors (e.g. tumor depth, degree of pigmentation of malignant cells, as well as skin phototype) can influence this complex biochemical reaction," you can express it more precisely, for example: "Various factors, including tumor depth, the degree of pigmentation in malignant cells, and the individual's skin phototype, can impact the outcome of this intricate biochemical process."
In the sentence "PDT was known to poorly work in pigmented lesions," you can phrase it more precisely, for example: "It has been widely recognized that PDT exhibits limited efficacy in the treatment of pigmented lesions."
Introduction:
In the sentence "the risk of local recurrence," it might be more precise to specify that it refers to "the risk of local tumor recurrence."
In the sentence "PDT can induce tumor death via direct induction of cell death," the phrase "direct induction of cell death" seems redundant, as PDT typically works by inducing processes like apoptosis, eliminating the need for a double use of "direct induction."
In the sentence "PDT can induce tumor death via direct induction of cell death, vascular damage, and activation of immune response," you could provide a more detailed discussion of the mechanisms through which PDT affects these processes.
In the sentence "different cell death pathways that mediate cytotoxicity of PDT may play a role, such as necrosis, apoptosis, and autophagy," you can mention more detailed mechanisms related to these pathways that may be involved in PDT's cytotoxicity.
Chapter 2:
In the sentence "Melanin is synthesized enzymatically at roughly 10-nm granular sites studding the internal walls of the organelles – melanosomes," you may want to clarify that it refers to "Melanin is enzymatically synthesized at approximately 10-nm granular sites that stud the internal walls of organelles known as melanosomes."
In the sentence "The average epidermal absorption coefficient (μa) depends on both, the melanosomal μa (1.7 x 1012 nm-3.48 [cm-1] for skin, where nm equals to wavelength in nm) and the volume fraction (fv) of melanosomes in the epidermis," you could express it more clearly as "The average epidermal absorption coefficient (μa) depends on both the melanosomal μa (1.7 x 1012 nm-3.48 [cm-1] for skin, where nm equals the wavelength in nm) and the volume fraction (fv) of melanosomes within the epidermis."
In the sentence "Moreover, the μa of sun-protected skin are generally lower than those of sun-exposed areas within each skin type group in the wavelength range from 500 to 900 nm," you should mention "the μa values" and correct it to "sun-protected skin is generally lower than that of sun-exposed areas within each skin type group in the wavelength range from 500 to 900 nm."
In the sentence "Photosensitizers used for PDT to treat dermatologic conditions such as 5-aminolevulinic acid (5-ALA) and its methyl ester (methyl aminolevulinate – MAL) are mainly metabolized into protoporphyrin IX (PpIX), which has five absorption wavelength peaks," you can add "are metabolized" for greater precision, like this: "Photosensitizers used for PDT to treat dermatologic conditions, such as 5-aminolevulinic acid (5-ALA) and its methyl ester (methyl aminolevulinate – MAL), are mainly metabolized into protoporphyrin IX (PpIX), which has five absorption wavelength peaks."
In the sentence "This phenomenon is probably caused by oxidative stress generated by chromophores in the skin that absorb strongly in this region, such as melanins, PpIX, and β-carotene," adding "is likely" before "caused" would emphasize that it's a hypothesis, like this: "This phenomenon is likely caused by oxidative stress generated by chromophores in the skin that absorb strongly in this region, such as melanins, PpIX, and β-carotene."
In the sentence "Melanogenesis can be also inhibited via paracrine effect, when melanocytes are exposed to PDT-treated keratinocytes or dermal fibroblasts," you can add "can also be" before "inhibited" to express the idea more precisely, like this: "Melanogenesis can also be inhibited via a paracrine effect when melanocytes are exposed to PDT-treated keratinocytes or dermal fibroblasts."
Chapter 3:
In the sentence "The literature search followed PRISMA guidelines for systematic reviews and the Cochrane manual [55-56]," it's better to mention "The literature search followed PRISMA guidelines for systematic reviews [55] and the Cochrane manual [56]."
In the sentence "They suggested that the pigment within the pBCCs prevents adequate light absorption and hinders PDT effect in those lesions," you can specify that "They suggested that the pigment within the pBCCs may prevent adequate light absorption, hindering the PDT effect in those lesions."
In the sentence "Ramirez et al. treated sixteen pBCC lesions with MAL-PDT in two sessions," you can say "Ramirez et al. treated sixteen pBCC lesions with MAL-PDT in two treatment sessions" for clarity.
In the sentence "A 2-month course of oral etretinate 0.5 mg⁄kg per day was used to decrease superficial scaling," you can add "A 2-month course of oral etretinate at a dose of 0.5 mg⁄kg per day was used to decrease superficial scaling."
In the sentence "The Japanese group, led by Itoh, treated sixteen pBCCs (eleven nodular, two superficial, and three ulcerative) located on the head," you can add "The Japanese group, led by Itoh, treated sixteen pBCCs (eleven nodular, two superficial, and three ulcerative) located on the head and neck area" to provide a more precise description of the location.
In the sentence "This resulted in removal of visible pigmentation (partial or total) and reduced tumor volume," you can say "This resulted in the removal of visible pigmentation (either partially or completely) and reduced the tumor volume."
In the sentence "The clinical follow-up showed no recurrence for all lesions (mean time of follow-up 24 months)," you can mention "The clinical follow-up showed no recurrence for all lesions with a mean follow-up time of 24 months."
In the sentence "Sung et al. in their study ablated epidermis of three histologically confirmed pBCCs by a fractional CO2 laser," you can add "Sung et al. in their study ablated the epidermis of three histologically confirmed pBCCs using a fractional CO2 laser."
Chapters 4 and 5:
All good
Conclusions:
It's worth emphasizing the importance of research on new photosensitizers that specifically target pBCC cells and research on factors contributing to patient resistance to treatment.
Other:
Table 1 needs to be revised for better readability, as the current layout is too complex.
Adding 2 or 3 visual illustrations for selected chapters would enhance the reader's experience.
Comments on the Quality of English Language
The article is written in an understandable manner but contains some language errors and inconsistencies. Here are some suggestions to improve the article's quality:
There are minor punctuation errors in a few places, such as missing commas before "and" or "or," which can create ambiguity in sentences.
In some places, the sentences are slightly convoluted and could be made clearer. Strive for greater clarity in conveying ideas.
In some instances, complex terminology is used that may be challenging for readers outside the field of medicine. Aim for simpler and more accessible language, especially in scientific articles.
Some sentences need restructuring to be more understandable.
Overall, the article is understandable, but some improvements could enhance its ability to convey information to the reader.
Author Response
We are grateful to the reviewer for his very supportive comments and for spending time reviewing our manuscript. We have corrected all mistakes as indicated by the reviewer. We have also modified Table 1 and added new figures. We hope that these modifications will satisfy the reviewer.
Round 2
Reviewer 1 Report
Comments and Suggestions for Authors
The authors have adequately responded to my comments.
Reviewer 2 Report
Comments and Suggestions for Authors
I accept in the present form.